# Airway management during ongoing chest compressions–direct vs. video laryngoscopy. A randomised manikin study

Richard Steffen[1,2], Simon Hischier[1], Fredy-Michel Roten[1], Markus Huber[1], Jürgen Knapp[1,3,4]*

1 Department of Anaesthesiology and Pain Medicine, Inselspital, Bern University Hospital, University of Bern, Bern, Switzerland, 2 Department of Intensive Care Medicine, Inselspital, Bern University Hospital, University of Bern, Bern, Switzerland, 3 Department of Anaesthesiology, Intensive Care Medicine and Pain Medicine, Hospital of Schwyz, Schwyz, Switzerland, 4 Department of Emergency Medicine, Inselspital, Bern University Hospital, University of Bern, Bern, Switzerland

* juergen.knapp3@googlemail.com

## Abstract

### Background

Tracheal intubation is used for advanced airway management during cardiac arrest, particularly when basic airway techniques cannot ensure adequate ventilation. However, minimizing interruptions of chest compressions is of high priority. Video laryngoscopy has been shown to improve the first-pass success rate for tracheal intubation in emergency airway management. We aimed to compare first-pass success rate and time to successful intubation during uninterrupted chest compression using video laryngoscopy and direct laryngoscopy.

### Methods

A total of 28 anaesthetists and 28 anaesthesia nurses with varied clinical and anaesthesiological experience were recruited for the study. All participants performed a tracheal intubation on a manikin simulator during ongoing chest compressions by a mechanical resuscitation device. Stratified randomisation (physicians/nurses) was performed, with one group using direct laryngoscopy and the other using video laryngoscopy.

### Results

First-pass success rate was 100% (95% CI: 87.9% - 100.0%) in the video laryngoscopy group and 67.8% (95% CI: 49.3% - 82.1%) in the direct laryngoscopy group [difference: 32.2% (95% CI: 17.8% - 50.8%), p<0.001]. The median time for intubation was 27.5 seconds (IQR: 21.8–31.0 seconds) in the video laryngoscopy group and 30.0 seconds (IQR: 26.5–36.5 seconds) in the direct laryngoscopy group (p = 0.019).

**Data Availability Statement:** All relevant data are within the paper and its Supporting Information file.

**Funding:** The authors received no specific funding for this work.

**Competing interests:** The authors have declared that no competing interests exist.

**Abbreviations:** CI, confidence interval; CPR, cardiopulmonary resuscitation; DL, direct laryngoscopy; FPS, first pass success; IQR, interquartile range; ROSC, restoration of spontaneous circulation; SMD, standardised mean difference; TTI, time to tracheal intubation; VL, video laryngoscopy.

## Conclusion

This manikin study on tracheal intubation during ongoing chest compressions demonstrates that video laryngoscopy had a higher first-pass success rate and shorter time to successful intubation compared to direct laryngoscopy. Experience in airway management and professional group were not significant predictors. A clinical randomized controlled trial appears worthwhile.

## Background

The paramount importance of providing continuous chest compressions with the shortest possible interruption in the context of cardiopulmonary resuscitation (CPR) has already been proven to be prognostically relevant [1], is highlighted accordingly in the corresponding resuscitation guidelines [2], and undisputed. However, regarding airway management during CPR there has been much discussion in recent years. While several studies suggest tracheal intubation as the preferred option for airway management during CPR [3–6], randomised controlled trials found no benefit in relevant outcome parameters for patients with tracheal intubation compared to supraglottic airway management [7–9]. One reason for this might be interruptions of chest compressions during laryngoscopy and tracheal intubation [10–12]. At best, intubation should be performed with ongoing chest compressions, thereby eliminating the associated no-flow time.

In recent years, video laryngoscopy (VL) has become established in everyday elective anaesthesia as well as in emergency situations and in the preclinical setting. In comparison to direct laryngoscopy (DL), a significantly better view of the vocal cord level was demonstrated, even in the presence of difficult airway anatomy [13, 14]. These findings were also confirmed in everyday preclinical work [15]. In addition, an increased first-pass success (FPS) rate in emergency situations has been shown [16–19]. This effect was demonstrated in inexperienced users in particular [20–22]. Overall, the data situation remains inconclusive. However, we hypothesize that VL facilitates safe and expeditious tracheal intubation even under the aggravated conditions of ongoing chest compressions. To investigate this, in this randomized controlled study we recorded FPS rate and time to successful intubation with VL and compared them to DL in an adult manikin model with uninterrupted chest compressions.

## Materials and methods

In December 2021, 28 anaesthesiologists and 28 anaesthesia nurses were recruited from the University Clinic for Anaesthesiology and Pain Medicine at the Inselspital in Bern, Switzerland. Participation was voluntary and free of charge. Written consent to use the data was obtained from all participants and they were assured that performance in the study scenario had no influence on the course result or any professional consequences. Subjects were blinded to the endpoints: they were only told that the study wanted to investigate whether endotracheal intubation could be performed "quickly and safely" with ongoing resuscitation. The participants' clinical experience in anaesthesiology ranged from less than one to 34 years. All participants had extensive training in both DL and VL. Before the start of the study, the cantonal ethics committee of Bern confirmed that no authorization was required (Req-2021-01338, 11/22/2021). This study was designed as a pragmatic pilot study, and participants were recruited as part of an in-hospital training course. Thus, the size of the group, the profession of the

subjects, and also their clinical experience were predetermined. A power analysis was deliberately omitted due to this setting. Nonetheless, effect sizes were statistically examined during the analysis.

## Study protocol

The study was carried out using a patient manikin (Resusci Anne Advanced SkillTrainer®, Laerdal Medical, Norway). These simulators are regularly used at our clinic for basic life support training and all participants have trained with this simulator before, but did not intubate this manikin before. There was no additional artificial aggravation of the airway anatomy. The manikin was placed dorsally on a standard table, while the chest compressions were performed using a mechanical resuscitation device during intubation (Corpuls CPR®, GS Elektromedizinische Geräte, Germany). In order to simulate more realistic movement of the torso during intubation, the manikin was elastically supported. The process of the intubation was documented by video recording. Participants were instructed to perform tracheal intubation with ongoing chest compressions. In the event of failed intubation, a new attempt had to be made.

## Randomisation and material

Stratified randomisation (physicians/nurses) was carried out using prepared and sealed envelopes in two groups. One group used VL (C-MAC®, Storz, Germany) with a Macintosh blade size 4, and the control group used DL with a Macintosh blade size 4. The groups used identical materials: VL (with external monitor), endotracheal tube size 7 with flexible tip and stylet (Parker Flex-Tip tube, Parker, USA) and ventilation bag (Spur II, Ambu, Germany). We deliberately chose to use a laryngoscope with an external monitor, although in everyday preclinical practice a device with a monitor on the handle is more likely to be used. This choice allowed standardisation for both groups (identical spatula and handle) and also allowed identical video analysis of the successful passage of the vocal cords for both groups. In the DL group, the monitor was swivelled out of the subject's field of vision. All other conditions remained identical to the VL group. The manikin airway, tube and stylet were lubricated beforehand.

After the conditions were explained and understood by the participant, the start of action was marked with a visual and acoustic countdown. Stopping chest compressions for intubation was not allowed. The study was carried out in a secluded room, and the study staff were not involved in the training of the individual study participants. To avoid any teaching bias, all participants underwent individual evaluation and other study participants were not allowed to watch.

## Endpoints

FPS rate was taken as the primary endpoint: Any intubation in which the tracheal intubation succeeded on the first attempt was interpreted as FPS. If there was oesophageal misalignment during intubation, that intubation was rated as a failure. Time to intubation (TTI) was assessed as the second endpoint: The above-mentioned countdown for the start of the study intervention was used as the starting point. The time was stopped as soon as the first effective ventilation was performed with the ventilation bag via the tracheal tube. This was confirmed by means of video analysis. For this purpose, the recordings of a camera in the room as well as the recording of the video laryngoscope were used. Furthermore, the first successful ventilation was verified using the manikin's internal detection system. An external examiner who was not involved in the study gathered data on FPS rate and TTI.

## Statistical analysis

Descriptive statistics are presented with counts and frequencies for categorical variables and with median and interquartile range (IQR) for continuous variables. Uncertainty inferences [95% confidence intervals (95% CI)] for proportions and their group differences (i.e., for the primary endpoint) are based on Wilson's confidence interval and score interval for difference of proportions. The group differences in the primary endpoint (FPS) were analysed with a one-sided Fisher's exact test, whereas the group difference for the secondary endpoint (TTI) was investigated with a one-sided Wilcoxon rank sum test. Predictors for the primary endpoint were investigated with a multivariable logistic regression model, where the goodness-of fit was examined using several pseudo-$R^2$ measures (McFadden, Cox and Snell, Nagelkerke) and the Brier-Score. A p-value $<0.05$ was considered statistically significant.

All calculations were performed using R version 4.0.2 [23], in particular with the packages *PropCIs* for confidence intervals [24], *rcompanion* for the goodness-of-fit measures [25] and *DescTools* for the computation of the Brier-Score [26].

## Results

Results of all 28 anaesthetists and 28 anaesthesia nurses were included in the study. The participants' experience in anaesthesia ranged from 0.75 to 34 years. The subjects were randomised into the two groups VL and DL, each with 28 participants (Table 1).

FPS was 100% (95% CI: 87.9% - 100.0%) in the VL group and 67.8% (95% CI: 49.3% - 82.1%) in the DL group [group difference: 32.2% (95% CI: 17.8% - 50.8%), p<0.001, Table 2]. The median TTI was 27.5 seconds (IQR: 21.8–31.0 seconds) in the VL group and 30.0 seconds (IQR: 26.5–36.5 seconds) in the DL group (p = 0.019, Table 3).

We performed a multivariable logistic regression analysis of FPS rate to assess the influence of experience or occupational group. Here, experience was not a significant predictor (p = 0.6). Likewise, no significant difference between the occupational groups could be demonstrated (p = 0.4, Table 4).

## Discussion

In our randomized manikin study, we were able to show that during ongoing chest compressions a higher FPS rate can be achieved and that the time until successful intubation is significantly shorter with VL compared to DL.

To improve the chances of a favourable neurological outcome for patients after cardiac arrest, every effort has to be made not only to achieve the highest possible FPS [27], but also to optimize circulation during CPR with as little no-flow time as possible. Airway management during CPR has been a highly discussed and intensively investigated research area during the last years. Three studies suggest that tracheal intubation is associated with higher short- and

**Table 1. Participants' characteristics.**

| | direct laryngoscopy (DL) | video laryngoscopy (VL) | SMD[1] |
|---|---|---|---|
| | N = 28 | N = 28 | |
| **Profession**: | | | 0.143 |
| Physician | 13 (46.4%) | 15 (53.6%) | |
| Nurse | 15 (53.6%) | 13 (46.4%) | |
| **Experience** (yrs) | 9.00 [5.00;19.2] | 5.00 [3.00;10.0] | 0.411 |

[1]SMD = Standardized Mean Difference

**Table 2. Primary outcome: First-pass success presented with counts and percentages.**

|  | direct laryngoscopy (DL) | video laryngoscopy (VL) | P |
|---|---|---|---|
|  | N = 28 | N = 28 |  |
| **First Pass Success (FPS)** |  |  | 0.0009[†] |
| Unsuccessful | 9 (32.1%) | 0 (0.00%) |  |
| Successful | 19 (67.9%) | 28 (100%) |  |

[†] Fisher's Exact Test for Count Data

long-term survival rates and better neurological recovery [4–6]. One large randomized controlled trial showed no significant difference in favourable functional outcome 30 days after OHCA between patients ventilated by a supraglottic airway compared with a tracheal tube [9]. Wang et al. [7] concluded from their PART trial that a strategy of initial laryngeal tube insertion was associated with significantly greater 72-hour survival compared with a strategy of initial tracheal intubation. However, this study has the significant limitation that the first pass success rate in patients who were tracheally intubated was only 51%. Therefore, a possible explanation for the inconclusive results of these studies on airway management might be the low success rates in tracheal intubation and interruptions of chest compressions for this procedure in contrast to positioning of an SGA. Future research on airway management during CPR should focus on how to improve FPS and reduce no-flow time for tracheal intubation.

According to our results, VL for airway management during CPR allows tracheal intubation with an excellent FPS even during uninterrupted chest compressions and performs significantly better than DL. The practicability and superiority of the VL—even with difficult airway anatomy, in emergency situations as well as complex preclinical situations—have been shown in several studies [16, 18, 19]. In addition, VL is also a suitable technique for inexperienced users to establish a safe airway in an emergency situation [28, 29]. This could even be demonstrated in beginners in the hospital setting [20–22]. Specifically, in a recent study of novice medical students, Keresztes and colleagues found that VL was superior to DL in terms of TTI in a normal airway and non-inferior in a difficult airway. In addition, a significantly smaller oesophageal false intubation rate could be observed in both cases [30].

Our regression analysis did not find an effect of either profession (physician, nurse) nor working experience in anaesthesia as a significant predictor of FPS. This result is interesting in that it could be postulated that users with many years of experience might be able to intubate more safely using DL. This could be assumed since these participants were regularly intubating patients using DL long before the introduction of VL. In addition to the proven advantage of VL for inexperienced users, it does not seem to be inferior for experienced practitioners either. Therefore, VL may provide further benefits in the prehospital setting, and—according to the finding of two recent studies—especially for providers with little experience in tracheal intubation [18, 19].

**Table 3. Secondary outcome: Time to intubate–only in the case of FPS.**

|  | direct laryngoscopy (DL) | video laryngoscopy (VL) | p |
|---|---|---|---|
|  | N = 19 | N = 28 |  |
| **time to intubation (TTI)** (median time in seconds) | 30.0 [26.5;36.5] | 27.5 [21.8;31.0] | 0.019[†] |

[†] Wilcoxon rank sum test

**Table 4. Multivariable regression for the outcome first-pass success.**

| Predictor | OR[1] | 95% CI[1] | p |
|---|---|---|---|
| **Experience** (yrs) | 0.98 | 0.91, 1.06 | 0.64 |
| **Profession** | | | |
| Physician | ref. | ref. | |
| Nurse | 0.51 | 0.09, 2.55 | 0.41 |

[1]OR = Odds Ratio, CI = Confidence Interval

Pseudo-$R^2$: 0.029 (McFadden), 0.025 (Cox and Snell), 0.043 (Nagelkerke). Brier-Score: 0.13

Although VL is becoming increasingly widespread it cannot be assumed that this technology is available everywhere. Nonetheless, for example, in German HEMS and the Swiss Air-Rescue (REGA) VL is used as the first-line device [13, 18]. Of course, the significantly higher costs for acquisition and maintenance compared to a conventional laryngoscope need to be noted. However, the trend here is towards falling prices and thus increasing availability [31].

Our study has clear limitations. Firstly, one issue is the use of a manikin that can only partially depict anatomical conditions. However, the benefit of using manikins is that we were able to establish standardized airway conditions for each participant. Secondly, due to the fact that the subjects were only present for one day, a cross-over design was not possible. In the setting of a cross-over design, the two intubations would have taken place within a short period of time and we then would expect a significant bias due to the short-term experience and training effect. This would not fulfil our goal of conducting a pragmatic trial simulating the setting of prehospital emergency care medicine where personnel are regularly confronted with unfamiliar situations for tracheal intubation. Thirdly, as this study was designed as a pragmatic pilot study, we could not complete a power analysis given the fixed population size. In this context, the small sample group must also be noted, despite the proven effect size. Fourthly, the clinical relevance of the absolute difference of 2.5 seconds in TTI can be questioned. For this reason, we chose FPS as the primary endpoint, which seems to be outcome relevant in CPR [27]. Fifthly, we did not include the experience with tracheal intubation as potential confounder for our multivariable regression model. However, this variable would very closely correlate with working experience in anaesthesia in our clinic at a university hospital. Therefore, we constrained to the working experience in anaesthesia as a simply and objectively measurable parameter.

## Conclusions

In summary, the results of our study provide a good basis for a large-scaled clinical randomized controlled trial comparing VL and DL under ongoing chest compressions in CPR. We were able to show significantly higher FPS for VL under these circumstances as compared to DL. Airway management experience and professional group were not significant predictors of FPS.

## Supporting information

**S1 Data.**
(XLSX)

## Acknowledgments

The authors would like to thank the volunteers for participating in this study, as well as the Bern Simulation- and CPR-Centre (BeSiC) at the Bern University Hospital for providing

facilities and equipment. We also would like to thank Jeannie Wurz, Medical Editor, and Dar-ren Hight, PhD, Bern, Switzerland, for their editorial assistance.

## Author Contributions

**Conceptualization:** Richard Steffen, Simon Hischier, Fredy-Michel Roten, Jürgen Knapp.

**Data curation:** Simon Hischier.

**Formal analysis:** Richard Steffen, Markus Huber.

**Writing – original draft:** Richard Steffen, Markus Huber, Jürgen Knapp.

**Writing – review & editing:** Fredy-Michel Roten, Jürgen Knapp.

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
