## [Decision Letter · Decision Letter 0]

27 Sep 2022

PONE-D-22-20206Airway management during ongoing chest compressions – direct vs. video laryngoscopy. A randomized manikin study.PLOS ONE

Dear Dr. Knapp,

Thank you for submitting your manuscript to PLOS ONE. After careful consideration, we feel that it has merit but does not fully meet PLOS ONE’s publication criteria as it currently stands. Therefore, we invite you to submit a revised version of the manuscript that addresses the points raised during the review process.

We look forward to receiving your revised manuscript.

Kind regards,

Billy Morara Tsima, MD MSc

Academic Editor

PLOS ONE

Journal Requirements:

Additional Editor Comments:

While these are interesting findings based on limited data from an in-house manikin study, they add important knowledge to the science. Overall, the results are presented well and the manuscript reads well. The discussion has not highlighted the important limitation of the small sample size and lack of power calculations associated with the study. There is limited discussion with regard to the multivariable logistic regression modeling used in the study. An important limitation of the model is that other potential confounders were not included to fit the model as these were not measured but could have been measured. This is worth highlighting in the discussion (e.g Age, experience with intubation-even though inferred by the variable "experience in anesthesiology"). Although goodness of fit assessment was done and the result presented with Table 4, there is no discussion of these results to assist readers in understanding how well the data fit the model.

Reviewers' comments:

Reviewer's Responses to Questions

**Comments to the Author**

1. Is the manuscript technically sound, and do the data support the conclusions?

Reviewer #1: Partly

Reviewer #2: Yes

Reviewer #3: Partly

2. Has the statistical analysis been performed appropriately and rigorously? 

Reviewer #1: Yes

Reviewer #2: I Don't Know

Reviewer #3: I Don't Know

3. Have the authors made all data underlying the findings in their manuscript fully available?

Reviewer #1: Yes

Reviewer #2: Yes

Reviewer #3: No

4. Is the manuscript presented in an intelligible fashion and written in standard English?

Reviewer #1: Yes

Reviewer #2: Yes

Reviewer #3: No

5. Review Comments to the Author

Reviewer #1: Thank you for the opportunity to review the presented manuscript entitled “Airway management during ongoing chest compressions – direct vs. video laryngoscopy. A randomized manikin study“. Authors showed higher FPS rate and shorter time to successful intubation with video laryngoscopy. The study is interesting and the manuscript well written. However, there are a few issues that need to be solved.

1. Authors state tracheal intubation is the gold standard during CPR. This is not true. Bag mask ventilation has the same weight/importance.

2. The FPS was higher in the VL group and the TTI was shorter. However, the difference in TTI was only 2.5 seconds. This seems to be clinically insignificant, especially during CPR (at least in non-hypoxic causes). Moreover, if the 9 unsuccessful first intubation attempts were also counted in the TTI it further diminishes importance of these findings.

3. As stated by the authors, crossover design would be much more appropriate.

4. Do I understand it right, that a VL with external monitor was used? In this setting (ongoing CPR) a “screen on the handle” device would be more appropriate (transport reasons). From a clinical practice, it seems to me, that using the “screen on the handle” device might be more difficult than using the external monitor.

5. Did all of the participants intubate manikin before? Or were there differences?

6. Statement in the discussion that VL should be recommended for tracheal intubation during CPR is too strong and not supported.

7. A chapter alone are SGA devices. For an inexperienced rescuer, the use of SGA might be easier, faster and also without chest compression interruptions.

Reviewer #2: This paper reports a randomised study comparing direct laryngoscopy (DL) to videolaryngoscopy (VL) in a scenario using a mannikin undergoing mechanical chest compressions. The primary endpoint is successful intubation at the first attempt (%) with a secondary end point of time to intubation (median in seconds).

Overall impression - simple study and methodology. Useful in proving utility of VL in achieving first pass successful intubation in a low fidelity mannikin scenario without interrupting chest compressions. This may have some translatable utility in real life resuscitation scenarios. The report is clear and easy to understand and the conclusions drawn are reasonable. In my opinion there are only minor issues (included in the more detailed review below).

The title is clear and succinct. It includes the information that this is a randomised trial. The question is relevant and the methodology seems appropriate.

The suggested keywords are all appropriate (although airway management is spelt wrongly). Others could be added e.g. "cardiac arrest" & "laryngoscopy" and I would suggest this paper is of interest to practitioners in anaesthesia / anaesthesiology and intensive care / critical care in addition to emergency medicine.

The abstract is easy to understand and contains the key information from the study. My only reservation is with the assertion that intubation is the "gold standard". There is some work demonstrating that laryngeal mask airways are not inferior for airway management during cardiopulmonary resuscitation with ongoing chest compressions. Your target audience will be aware of this and might prefer a more measured statement e.g. "some patients require intubation as part of their cardiopulmonary resuscitation" or similar. I do note that you have covered this in your introduction with references to support endotracheal intubation but I think it would be reasonable to acknowledge that there is still controversy. This is an opinion rather than a barrier to publication.

The ethics approval is described and it is confirmed that written consent was obtained from participants (which seems appropriate). Unclear what information was given to potential participants and which potential issues were anticipated e.g. what would happen if a participant was concerned after a "poor performance".

The complete data is included in a link that opens an Excel table that is easy to access.

The introduction is short but does explain the background to the study. There are appropriate references. However, there are studies on this topic (VL vs DL during CPR) in real life resuscitation scenarios (e.g. Lee et al., 2015; Park et al., 2015) which have not been referenced. Other studies have only been referred to very briefly. For example, the study by Risse et al. (2020) is referenced but addresses a more similar question than is implied. These studies have typically been observational in nature and not randomised. There are also some randomised studies comparing DL to VL for intubation during CPR (Kim et al., 2016). In addition there are other mannikin / simulation studies addressing this question, including Keresztes et al. who published in this journal in November 2021. It would help if the authors explained how their study relates to these studies. Why have you chosen your methodology to answer the question? What does this study add? Does it tell us something new or does it reinforce the findings of these studies in a different setting or with different equipment or participants? Also, somewhere in the introduction you could clarify that the population considered are adult patients only.

The materials and methods section is the bulk of the report and is detailed. I think it would be possible to replicate this study reasonably closely with the information provided. We wondered about a few details e.g. Is there any reason for the number of anaesthetists and anaesthetic nurses included? How did the investigators decide on this? Why were these practitioners chosen (as opposed to say intensivists or paramedics)? Was there a standardised script used for the participants? Were participants blinded to the primary and secondary endpoints? Presumably the answer to the first 3 questions is that it was a pragmatic methodology but it might be reasonable to say this.

There is detail on the statistical analyses used (and links to the programs in the references).

The results section is short but conveys the important results. There are no tables or charts in the body of the report. The average time to intubation and the number of attempts required when there was more than one attempt at intubation are not included. It may be useful to demonstrate the range of intubation attempts in the DL group (especially given 32% in the DL group had to attempt reintubation).

The discussion is also short and could include more detail on why first pass intubation and rapid intubation during cardiopulmonary resuscitation are of benefit. It would be reasonable to explain the benefits to patients and practitioners in the range of settings in which patients in cardiac arrest are encountered. You have addressed some limitations to the study but what about downsides to universal videolaryngoscopy? You may not feel there are any significant barriers but this should be explained as readers may not agree. Examples might be cost (particularly in resource limited settings) or portability or maintenance. I note that the methods sections describes the CMAC videolaryngoscope with the screen turned away for the direct laryngoscopy attempts - this indicates that it was not a portable videolaryngoscope used in the study. I don't think this detracts from the findings but could be considered in the discussion.

The conclusions are reasonable (although they might feel more honest without the intensifiers e.g "support" rather than "strongly support").

We had a few questions about the tables. These are listed below:

Table 1- what does SMD stand for.

Table 2 - line 182 includes a spelling mistake - Fist should be FIRST.

Table 3 - can "seconds" be changed to "median time in seconds" (for clarity and easy reading).

Comments on competing interests, funding, acknowledgements and authors' contributions are all included.

The references are consistent in style and the links work.

Studies referred to above:

Keresztes D, Mérei A, Rozanovic M. Nagy E, Kovács-Ábrahám Z, Oláh J, Maróti P, Rendeki S Nagy B & Woth G. Comparison of VividTrac, King Vision and Macintosh laryngoscopes in normal and difficult airways during simulated cardiopulmonary resuscitation among novices. PLoS One. 2021;16(11):e0260140. doi: 10.1371/journal.pone.0260140.

Kim JW, Park SO, Lee KR, Hong DY bake KJ, Lee YH, Lee JH & Choi PC. Video laryngoscopy vs. direct laryngoscopy: Which should be chosen for endotracheal intubation during cardiopulmonary resuscitation? A prospective randomized controlled study of experienced intubators. Resuscitation. 2016;105:196-202. doi: 10.1016/j.resuscitation.2016.04.003.

Lee DH, Han M, An JY, Jung JY, Koh Y, Lim C, Huh JW & Hong S. Video laryngoscopy versus direct laryngoscopy for tracheal intubation during in-hospital cardiopulmonary resuscitation. Resuscitation. 2015;89:195-9. doi: 10.1016/j.resuscitation.2014.11.030.

Park SO, Kim JW, Na JH, Lee KH, Lee KR, Hong DY & Baek KJ. Video laryngoscopy improves the first-attempt success in endotracheal intubation during cardiopulmonary resuscitation among novice physicians. Resuscitation. 2015;89:188-94. doi: 10.1016/j.resuscitation.2014.12.010

Risse J, Volberg C, Kratz T et al. Comparison of videolaryngoscopy and direct laryngoscopy by German paramedics during out-of-hospital cardiopulmonary resuscitation; an observational prospective study. BMC Emergency Medicine. 2020; 20: 22. doi.org/10.1186/s12873-020-00316-z

Reviewer #3: Comments to the authors:

1. I agree that VL have been shown to provide better glottic view, but not necessarily higher intubation success as well as fastest intubation compared to DL in all cases. It does help in patients with difficult intubations, but not in cases not having any difficulty. Moreover, the success as well as first pass rate varies with different types of DLs. You have not touched this area. Recent meta-analysis on DL including Cochrane one does not echo your whole positive statement on VL compared to DL.

2. Why participants’ experience so varied? There should have been some standardization. Did you do power analysis as well as subgroup analysis on different level of experiences and categories of those did intubation or not?

3. English language used here needs some minor correction as well as editing. Mixture of both UK and USA version of English has been used here.

4. I am bit confused about blinding. Though I can understand the randomization, but to my knowledge blinding is difficult between VL and DL.

5. How did you blind the assessors?

6. How did you make sure ventilation was effective? Bag ventilation as a method to confirm tracheal intubation is very poor confirmatory test in vivo.

7. Don’t repeat data of methodology in the result section.

8. Can you tell me how you have distributed or rather randomised the 28 anaesthetist and nurses? Also, how did you spread their years of experience in two groups?

9. I can see FPS was 100% in VL which is not even found in human trials, neither in published mannequin trial. I am surprised. This raises suspicion on randomization, blinding and assessment methods. I shall like to know whether you have used normal airway or simulated a difficult airway scenario here?

10. The TTI in DL has been proved beyond doubt to be much less than VL, which is opposite here. Can you explain the discrepancy?

11. Discussion should not start with aim which should be mentioned in the introductory section, rather discussion should start with most important positive and negative findings.

12. ETI is not recommended as first line of important part of management in CPR unless the oxygenation or ventilation is an issue. Airway can even be managed with supraglottic devices in case airway control is needed. ETI or advanced airway is only recommended after 2-3 cycles, if at all necessary, as per current ACLS protocol. I do not agree with your statement regarding securing airway at the earliest.

13. VL is not easily available and readily set up in emergency situation, rather it has utility in elective difficult airway scenario. Setting a VL takes time and manpower which is not practical in most cases. Also, the efficacy in struggling CPR time in real life scenario is not proven with VL, where DL is still gold-standard. I refuse to accept your concluding statement here.

14. Success of securing airway in critically ill lifesaving urgent set ups should not be assessed using beginners or newcomers in airway management. This is poor planning.

15. Many other limitations not mentioned including flaws in design, conduct, as well as data analysis.

Strengths:

• Idea is noble one to assess the utility of VL in CPR scenarios.

• No other major strengths can be seen here.

Weakness: Many to even mention.

• Serious flaws in study methods, randomization, blinding, assessment.

• Extrapolation of mannequin experience on human subjects is far from reality.

• The results look very much manipulated and suspicious to even believe.

• Defending the study result is equally poorly done.

Areas of improvements:

• English language.

• Flaws in methodology, randomisation and blinding should be clarified.

• The discrepancy in results should be explained compared to previously published reports.

• Literature review should be targeted one than mere descriptive one.

Decision: The topic chosen is a valid and timely one. But the current study is plagued by several flaws including design, methodology, outcomes. Considering the importance of the topic, I shall still like to give a chance to the authors to defend their side. If they come up with point-by-point clarification of my questions, I can give another look.

6. PLOS authors have the option to publish the peer review history of their article (what does this mean?). If published, this will include your full peer review and any attached files.

Reviewer #1: No

Reviewer #2: **Yes: **Katherine Chatten

Reviewer #3: No

---

## [Author Response · Author response to Decision Letter 0]

9 Dec 2022

Please see our attatched file "Response to reviewers" (File name: Revision PLOS ONE_clean.docx)

---

## [Editor Report · Decision Letter 1]

18 Jan 2023

Airway management during ongoing chest compressions – direct vs. video laryngoscopy. A randomised manikin study

PONE-D-22-20206R1

Dear Dr. Knapp,

We’re pleased to inform you that your manuscript has been judged scientifically suitable for publication and will be formally accepted for publication once it meets all outstanding technical requirements.

Kind regards,

Billy Morara Tsima, MD MSc

Academic Editor

PLOS ONE
---

## [Editor Report · Acceptance letter]

30 Jan 2023

PONE-D-22-20206R1 

Airway management during ongoing chest compressions – direct vs. video laryngoscopy. A randomised manikin study 

Dear Dr. Knapp:

I'm pleased to inform you that your manuscript has been deemed suitable for publication in PLOS ONE. Congratulations! Your manuscript is now with our production department. 

Kind regards, 

on behalf of

Dr. Billy Morara Tsima 

Academic Editor

PLOS ONE